# The New Aristocrat of Wuyi Rock Tea: Chemical Basis of the Unique Aroma Quality of “Laocong Shuixian”

**DOI:** 10.3390/foods14101706

**Published:** 2025-05-12

**Authors:** Yucheng Zheng, Yuping Zhang, Xiaoxi Ou, Qiuming Li, Huiqing Huang, Jianming Zhang, Feiquan Wang, Yutao Shi, Zhilong Hao, Bo Zhang, Yun Sun

**Affiliations:** 1College of Tea and Food Sciences, Tea Engineering Research Center of Fujian Higher Education, Tea Science Research Institute of Wuyi University, Wuyi University, Wuyishan 354300, China; 18094159524@163.com (Y.Z.); zjm0308@163.com (J.Z.); wfq1982@wuyiu.edu.cn (F.W.); ytshi@wuyiu.edu.cn (Y.S.); 2Key Laboratory of Tea Science, College of Horticulture, Fujian Agriculture and Forestry University, Fuzhou 350007, China; zhangyuping_tea@163.com (Y.Z.); ouxiaoxi_66@163.com (X.O.); 15280856651@163.com (Q.L.); hhq9959@163.com (H.H.); haozhilong@126.com (Z.H.)

**Keywords:** Laocong Shuixian, oolong tea, odor-active compounds, GC–O–MS

## Abstract

Laocong Shuixian (LCSX), a premium Wuyi rock tea derived from aged Shuixian tea trees, is valued by consumers for its distinctive “Cong flavor”—a unique aroma profile characterized by woody, bamboo leaf, and glutinous rice notes. However, the chemical basis and underlying mechanisms of this unique aroma remain unclear. Here, we assessed and established a professional sensory evaluation panel using the PanelCheck software, with significant F-value levels >5% confirming the panel’s discriminative capacity for key “Cong flavor” attributes. Combining a literature review and sensory analysis, we identified the descriptive terms associated with the “Cong flavor” of LCSX. Gas chromatography–olfactometry–mass spectrometry (GC–O–MS) analysis revealed 36 key aroma-active compounds, among which theaspirone (OAV = 500.05, ACI = 37%, R_woody_ = 0.82), *δ*-decalactone (OAV = 65.6, ACI = 4.3%, R_woody_ = 0.77), and 2-acetylpyrrole (OAV = 163, ACI = 9%, R_rice_ = 0.74) were identified as the contributors to the woody and rice-like notes of LCSX based on odor activity values and correlation analyses. Molecular docking results demonstrated that these compounds spontaneously bind to multiple olfactory receptors, with binding affinity ≤−5.0 kcal/mol, providing insights into their roles in human aroma perception: theaspirone to OR8D1; δ-decalactone to OR1E2, OR5M3, OR7D4, OR7G1, OR8D1 and OR8G1; and 2-acetylpyrrole to OR1E2, OR1G1, OR5M3, OR7D4, OR7G1, OR8D1, and OR8G1. This study enhances our understanding of the formation of distinctive aroma qualities in oolong tea and establishes a foundation for further research into its sensory and chemical properties.

## 1. Introduction

Wuyi rock tea, recognized as one of China’s top ten famous teas, epitomizes northern Fujian’s oolong tea varieties. This category includes diverse types, such as Wuyi Shuixian and Wuyi Rougui, categorized based on the tea varieties utilized during harvesting and production [1]. Among these, Wuyi Shuixian, derived from the esteemed Fujian Shui Xian (*Camellia sinensis* ‘Fujian Shuixian’), exemplifies superior quality and is a flagship product of Wuyi rock tea, noted for its distinctive orchid aroma and unique notes of woodiness and bamboo leaf aroma, making it highly desirable for consumers.

As a prominent variety of Wuyi rock tea (WRT), Wuyi Shuixian has garnered significant research interest due to its distinctive flavor profile. Existing studies on WRT have systematically investigated quality formation mechanisms from multiple perspectives, including postharvest processing [2], storage duration [3], geographical origin [4], and roasting technology [5]. Specifically for Shuixian, extensive research has explored the key compounds underlying its aroma quality. Sensory evaluation and Headspace–Solid Phase Micro Extraction–Gas Chromatography–Mass Spectrometry (HS–SPME–GC–MS) analyses revealed that lower-grade Shui Xian tea exhibits reduced floral/fruity aromas (associated with linalool, indole, phenethyl alcohol, etc.) but enhanced roasted/sweet notes (linked to pyridine and 2,5-dimethylpyrazine) [6]. Additionally, dynamic aroma changes during processing have been studied using gas chromatography–mass spectrometry (GC–MS) and have identified alcohols, alkenes, and esters as primary volatile compounds. The roasting process enhances the roasted, floral, and woody aromas of the tea, with 2-ethyl-3,5-dimethylpyrazine (odor activity value, OAV = 4.71) identified as a key contributor to the characteristic roasted fragrance [7]. Further aroma analyses of oolong teas from three varieties, Shuixian, Huang Meigui, and Zi Mudan, revealed that roasted, grainy, charred, woody, and floral notes are common flavor characteristics [8]. Key aromatic compounds such as 2-ethyl-3,5-dimethylpyrazine, linalool, and *β*-ionone were identified as shared components among Shuixian and other oolong tea varieties, and investigations into the aroma quality of Wuyi rock tea produced from 16 different varieties revealed that Shuixian tea has pronounced floral and woody aromas compared with those of other oolong teas [9]. Researchers have reported that the age of Shuixian tea trees significantly affects the aroma quality of the finished product. As these trees mature, distinctive aromas such as woodiness, bamboo leaf, and glutinous rice become more pronounced. This unique fragrance, termed the “Cong flavor” by tea professionals, is especially characteristic of Shuixian trees over 50 years old and is designated “Laocong Shuixian” (LCSX) [10]. Wuyi Shuixian tea, which has a pronounced “Cong flavor” aroma, is often associated with increased economic value. At present, many consumers are beginning to pursue Wuyi Shuixian products with a “Cong flavor”. Thus, investigating the key volatile compounds that influence the formation of the “Cong flavor” aroma is crucial for enhancing oolong tea aroma quality. However, research in this area remains limited.

Stir bar sorptive extraction (SBSE) is an innovative solid-phase microextraction technique that immerses a coated stir bar directly in a sample to adsorb volatile compounds [11]. Compared with traditional solid-phase microextraction methods, SBSE offers significant advantages, including higher adsorption capacity, increased enrichment factors, enhanced sensitivity, and improved reproducibility [12]. Most studies on the aroma quality of Wuyi Shuixian currently rely on conventional gas chromatography–mass spectrometry (GC–MS). Although GC–MS can identify aromatic components and their concentrations, it often fails to pinpoint the key volatile compounds responsible for aroma. Gas chromatography–olfactometry–mass spectrometry (GC–O–MS) effectively addresses this limitation by combining human sensory evaluation with GC–MS technology. This approach facilitates the rapid profiling and identification of aroma-active compounds and exploration of the relationships between odors and sensory characteristics, thereby elucidating the mechanisms underlying significant aroma formation [13]. The emerging field of molecular sensory science has further enhanced the application of GC–O–MS, which is increasingly utilized in aroma compound detection in oolong tea [14,15,16].

To systematically investigate the aroma characteristics of LCSX, a sensory evaluation panel was trained for this study. Quantitative descriptive analysis (QDA) was conducted on 12 collected LCSX samples. Key aromatic components were determined using SBSE–GC–O–MS technology, and OAVs and aroma character impact (ACI) values were used to identify the key aromatic compounds contributing to the distinctive aroma. Molecular docking was used to visualize and predict atomic-level interactions between olfactory receptors and key aroma-active substances. The aim of this research was to provide a theoretical basis for the high-quality processing of Shuixian oolong tea.

## 2. Materials and Methods

### 2.1. Samples and Reagents

Twelve samples of LCSX oolong tea were sourced from various tea companies in Mount Wuyi City, Fujian Province, to ensure representative selection. A letter from A to L was randomly selected as the sample number for each tea sample. The tea samples were immediately sealed and stored at 4 °C upon purchase until formal evaluation commenced. The n-alkanes ranging from C6 to C33 (chromatographic purity), 2-nonanol, and sodium chloride were obtained from Supelco (Bellefonte, PA, USA), Sigma (St. Louis, MO, USA), and China National Pharmaceutical Group Chemical Reagents Co., Ltd. (Shanghai, China), respectively.

### 2.2. Establishment, Training, and Performance Evaluation of the Sensory Panel

A sensory evaluation panel was formed by recruiting university students who held a Level 3 (senior) tea tasting certification. Panelists were selected based on the Chinese National Standard GB/T 16291.1-2012 “Sensory analysis—General guidance for the selection, training and monitoring of assessors—Part 1: Selected assessors” [17]. The initial screening involved aroma recognition tests and assessments of sample perception descriptions. Ultimately, six panelists were selected (two males and four females). The training and performance evaluation of the panel consisted of three phases. In Phase 1, the organizer provided three tea samples randomly to the panel members, who evaluated them following the methodology for sensory evaluation of tea GB/T 23776-2018 [18]. This phase yielded a set of aroma descriptors, leading to a consensus on the aromatic properties of LCSX based on the literature.

In accordance with our previously described method [19], in Phase 2, the QDA method was used to score the aroma characteristics of all 12 LCSX samples on a 10-point scale (0 to 10), where 0 = undetectable and 5 = extremely intense, with each sample evaluated twice. The scores were analyzed using the PanelCheck software (Version 1.4.2) to assess the panelists’ ability to differentiate samples, stability, and interpanelist consistency, thereby evaluating their sensory assessment capabilities and outcomes [20].

### 2.3. Determination of Volatile Components in LCSX

The instrumentation included an MPS Robotic Pro multifunctional automatic sampling system (Gerstel, Germany), a CIS4 large-volume cold trap injector (Gerstel, Germany), an ODP4 olfactometer (Gerstel, Germany), a TDU2 thermal desorption and adsorption stirrer (polydimethylsiloxane, 0.10 mm × 10 mm) (Gerstel, Germany), and a 7890A/5975C GC–MS system (Agilent Technologies, Santa Clara, CA, USA).

Our method was based on previous methods developed by our research group [21]. A total of 1.0 g of each sample was infused with 50 mL of purified water at 100 °C. After cooling to room temperature, 15.0 mL of the tea infusion were transferred to a 20 mL headspace vial, to which 40 μL of an internal standard (2-nonanol solution at 24 μg/mL) and 4.5 g of sodium chloride (NaCl) were added. A magnetic stirrer was then placed in the vial, and extraction was performed on a heating plate at 25 °C for 60 min with a stirring speed of 1200 r/min. Finally, the extracted stir bar was rinsed thoroughly with ultrapure water, wiped clean with lint-free tissue, and then placed in TDU2 for thermal desorption.

The temperature program was set to start at 40 °C for 2 min, followed by a ramp to 250 °C at a rate of 5 °C/min, with a hold time of 20 min. The carrier gas used was high-purity helium (99.999% or higher), with a flow rate of 1.8 mL/min. Injection was performed using a large-volume cold injection system, with the injector temperature set between −30 °C and 250 °C at 15 °C/s, in splitless mode. The thermal desorption temperature was set from 25 to 250 °C at a rate of 100 °C/min, also in splitless mode, and the transfer line temperature was maintained at 260 °C, with a split ratio of 1:1 for injection. The electron energy was set to 70 eV, with the ion source temperature, transfer line temperature, and quadrupole temperature set to 230 °C, 250 °C, and 150 °C, respectively. The mass scanning range was from 33 to 400 m/z, with an electrometric capacity of 1258 V.

Five members were randomly selected from our previously established sensory panel to participate in this olfactory experiment. The aroma intensity of each compound in the samples was evaluated via a 5-point scale. Panelists recorded the intensity and descriptors of the aroma compounds for each sample and engaged in discussions about the results. The intensity score for each sample was determined as the mean value calculated from the panel’s evaluations.

Qualitative identification was performed using the NIST 20 spectral library in conjunction with retention indices (RIs). Compounds with match quality values greater than 80 were selected for analysis. A mixture of n-alkanes was injected separately, following the same temperature program and preliminary GC–MS conditions. The RIs of the volatile compounds were calculated and compared with the values in the database. Using 2-nonanol as the internal standard, the concentrations of the components in the sample were determined based on the ratio of the peak areas of each component to the peak area of the internal standard.

### 2.4. Molecular Docking of the Binding Interactions Between the Aroma-Active Compounds and the Human Olfactory Receptors

Three-dimensional structural models of the aroma-active compounds and olfactory receptors were downloaded from the PubChem (https://pubchem.ncbi.nlm.nih.gov) (accessed on 13 December 2024) and UniProt (https://www.uniprot.org) (accessed on 13 December 2024) databases, respectively. Common human olfactory receptors used for subsequent molecular docking included human olfactory receptor 1E2 (OR1E2), human olfactory receptor 1G1 (OR1G1), human olfactory receptor 5M3 (OR5M3), human olfactory receptor 7D4 (OR7D4), human olfactory receptor 7G1 (OR7G1), human olfactory receptor 8D1 (OR8D1), and human olfactory receptor 8G1 (OR8G1). The specific sequence information of the olfactory receptors is listed in Appendix A. The aroma-active compounds (ligands) were hydrogenated and charged using the Chem3D software (Version 22.0.0). The PyMOL software (Version 3.0.5) was used to remove solvents and organics from the receptor proteins. The Autodock software (Version 1.5.7) was used to scan the docking sites between ligands and receptors, and the Vina software (Version 1.2.0) was used to calculate the docking results, keeping the parameters at their defaults [22]. Finally, the PyMOL software (Version 3.0.5) was used to visualize the molecular docking results.

### 2.5. Data Calculation and Analysis

The calculation formulas for OAV (1) and ACI (2) are as follows:(1)$$OAV=\frac{Cx}{OTx}$$(2)$$ACI(\%)=\frac{Ox}{\sum nOn}$$

In the above formulas, Cx represents the mass concentration of volatile compound x (in μg/L), OTx represents the odor threshold of volatile compound x in water (μg/L), Ox represents the OAV of volatile compound x, and ∑nOn represents the sum of OAVs for all key volatile compounds. All OAVs were derived using water-based odor thresholds.

One-way analysis of variance (ANOVA) was performed using SPSS (version 19.0; Chicago, IL, USA). Correlation analysis and visualization were conducted using the GraphPad Prism software (version 8.0; GraphPad Software, California, CA, USA). The sample differentiation capability, stability, and intermember consistency analysis and visualization were performed using the PanelCheck software (Version 1.4.2) [23], with the default parameters. Following previous methods [24], principal component analysis (PCA) and visualization were carried out in R Studio (Version 4.3.1) using the FactoMineR and Factoextra packages, also with default parameters.

## 3. Results

### 3.1. Sensory Evaluation Terminology and Selection of Group Members for LCSX Aroma

A panel of 10 trained sensory evaluators (five males and five females), all university students holding Level 3/Advanced tea tasting certifications, was recruited. Following the guidelines outlined in GB/T 16291.1—2012 for sensory analysis [17], the evaluators underwent aroma recognition tests and assessments of sample perception capabilities. Ultimately, two males and four females were selected as sensory panel members (designated WYU-1 to WYU-6).

We reviewed the previous literature on the aromatic characteristics of LCSX and summarized the findings, as shown in Table 1. A total of four studies described the “Cong flavor”, with all the evaluations indicating that it has a woody aroma. Some studies also noted the presence of additional notes, including bamboo leaf (or Indicalamus leaf/zongzi leaf) scent, mossy fragrance, floral undertones, and a slight brown rice aroma.

Furthermore, our sensory evaluation panel assessed 12 Laocong Shuixian samples based on GB/T 23776-2018 to obtain a more comprehensive and accurate evaluation [18] (Appendix A). Our evaluation revealed a total of six aroma descriptors for the LCSX. Among these aromas, floral, grassy, and sweet aromas do not correspond to the characteristic “Cong flavor” but are recognized as standard descriptors for oolong tea. Consistent with previous studies, we suggest that the “Cong flavor” is similar to woody and zongzi leaf aromas (Table 1). Notably, we did not identify any moss-like scents within the “Cong flavor”. Furthermore, our evaluation indicates that the “Cong flavor” is more akin to the scent of glutinous rice than to that of brown rice. The brown rice aroma reported in earlier studies is likely attributable to the roasting process or baking stage rather than the characteristic aroma of tea varieties [25].

**Table 1 foods-14-01706-t001:** Characteristics of the “Cong flavor” aroma quality of LCSX documented in the literature.

Descriptive Terms for “Cong Flavor” Aroma Quality	Author and Date
Woody aroma, bamboo leaf aroma, osmanthus or plum blossom aroma	(D. Chen et al., 2011) [26]
Woody aroma, zongzi leaf aroma, mossy aroma, orchid aroma, brown rice aroma	(Hong & Gong, 2020) [27]
Woody aroma, moss aroma, zongzi leaf aroma	(Shang et al., 2022) [28]
Floral aroma, woody aroma, “Cong flavor” (composite)	(F. Wang et al., 2020) [10]

Based on the sensory evaluation results and subsequent discussions, seven key aroma descriptors for LCSX were retained: floral, grassy, woody, glutinous rice, Indicalamus leaf, sweet, and the overall aroma intensity of the “Cong flavor” (Figure 1a). Among them, the aroma descriptors of floral, woody, and Indicalamus leaf fragrance were consistent with previous records. The sensory panel members evaluated the six aroma attributes of a total of 12 LCSX tea samples using a 10-point scale (the evaluation process was repeated twice, and the average score was taken), and the detailed scoring table is included in Appendix A. Based on the scores for the six aroma attributes of the 12 samples, we used the PanelCheck software (Version 1.4.2) to assess the sample differentiation capabilities and consistency of the sensory panel members.

As shown in Figure 1b, a higher F value indicates a stronger ability of evaluators to distinguish between different samples. The results revealed that the F values for the aroma attributes associated with the “Cong flavor” (woody, glutinous rice, Indicalamus leaf, and overall aroma) were all significant at the 5% level. Conversely, the F values for the common oolong tea aroma attributes, such as grassy and sweet notes, were below the 5% significance level, likely due to the less pronounced expression of these aromas among the 12 LCSX samples. Additionally, the clustering effects for the woody, glutinous rice, Indicalamus leaf, floral, and overall aroma points were the most pronounced in the boxplot, reaching highly significant levels, followed by grassy and sweet notes (not significant) (Figure 1c). These data suggest that, after training, the panel members exhibited good consistency and differentiation capabilities when the “Cong flavor” of the LCSX was evaluated, indicating that the group’s evaluation results were reliable.

### 3.2. SBSE–GC–MS Analysis of Four Representative LCSX Samples

Principal component analysis (PCA) is an effective method for calculating and visualizing the overall differences between samples. Therefore, we conducted PCA on the scoring matrix generated by the sensory evaluation panel for the 12 LCSX samples. The results are presented in Figure 2a, where principal component 1 (PC1, 65.9%) effectively separates the 12 samples into two distinct groups: one group includes samples B, H, L, E, and C, whereas the other consists of samples I, D, K, J, A, G, and F. Notably, samples H and L are the furthest from samples G and F, indicating significant differences in overall sensory evaluation between these groups. The radar plot of the six sensory attributes (Figure 2b) further supports these findings, showing that, in addition to the “grassy” and “sweet” attributes, there are notable differences in other aroma characteristics between samples H and L compared with samples G and F. This information can be used to better identify the key characteristic aroma components of LCSX. We selected samples H, L, G, and F for subsequent SBSE–GC–MS analysis. 

Through SBSE–GC–MS analysis, a total of 107 known volatile compounds were identified in the four LCSX samples. All measured information has been provided in Appendix A. These compounds primarily included alcohols, aldehydes, ketones, nitrogen-containing compounds, esters, phenols, and acids, with varying proportions of each volatile substance across different samples (Figure 2c). Notably, we observed that alcohols constituted the highest proportions in samples G and F, reaching 24.49% and 28.52%, respectively. In contrast, nitrogen-containing compounds were most abundant in samples H and L, accounting for 23.98% and 24.37%, respectively. This discrepancy likely reflects differences in metabolic pathways between the two groups, consistent with their significant sensory profiles. PCA of the volatile metabolite profiles of the LCSX samples revealed results that were consistent with the QDA scoring classification. Principal component 1 (Dim1, 49.4%) effectively distinguishes samples H and L from samples G and F (Figure 2d). Furthermore, we extracted the top ten volatile compounds contributing to principal component 1 (Dim1). These compounds, in order of contribution, were theaspirone, linalool oxide, dihydro-*β*-ionone, nerolidol, 2-furaldehyde, *γ*-hexalactone, acetylfuran, isovaleraldehyde, benzaldehyde, and linalool (Figure 2e). Among the top 10 compounds identified, some exhibited pronounced differences in concentration between the two groups of samples. For example, the levels of theaspirone and benzaldehyde were significantly greater in the H and L samples than in the G and F samples, whereas 2-furaldehyde demonstrated the opposite trend (Figure 2f). These volatile compounds likely play crucial roles in differentiating the key aromatic characteristics of the two groups of LCSX samples.

### 3.3. Identification of Four Representative Aroma-Active Components in LCSX via GC–MS

The odor activity value, which integrates the concentrations of aroma compounds with their respective thresholds, effectively reflects the contribution of each component to the overall aroma of the sample or specific aromatic properties. Generally, aroma compounds with OAVs > 1 are considered to contribute to the overall aroma profile of a sample, whereas those with OAVs > 10 significantly influence the overall fragrance. Accordingly, we consulted the relevant literature to determine the documented thresholds of aromatic components in water and subsequently calculated their corresponding OAVs and aroma contribution index (ACI) values. The results presented in Figure 3 indicate that a total of 20 aroma compounds presented OAVs > 1 across the four LCSX samples (detected in at least one sample), primarily consisting of terpenoids, aliphatic compounds, aromatic compounds, and lactones, among others, whereas 11 aroma compounds presented OAVs > 10 (also detected in at least one sample). The key compounds included linalool, jasmone, *β*-ionone, 3,5-octadien-2-one, *δ*-decalactone, 2-dimethylbutanal, trans-linalool oxide (pyranose), linalool oxide, (*E*)-2-heptenaldehyde, 2-acetylpyrrole, and theaspirone. Additionally, certain aroma compounds with relatively low odor activity values (OAVs)—such as phenylacetaldehyde, hexanal, octanal, indole, β-cyclocitral, and safranal—also contribute to shaping the overall aroma profile of LCSX. These findings highlight the critical roles of these compounds in shaping the aroma quality of LCSX.

In different samples, the same aromatic substances presented varying ACI values, which may primarily account for the observed aroma differences among the samples. For example, in sample H, the compound linalool oxide (OAV = 644.93, ACI = 37) contributed most significantly to the overall aroma, followed by theaspirone (OAV = 500.05, ACI = 29) and linalool (OAV = 158.84, ACI = 9). In sample F, 2-methylbutanal (OAV = 510.75, ACI = 27) was the predominant contributor to the overall aroma, followed closely by linalool oxide (OAV = 357.70, ACI = 19). Notably, in samples H and L, which exhibited prominent “Cong flavor”, the OAVs of typical floral and fruity aroma compounds such as 2,3-dimethylbutanal, *δ*-decalactone, phenylacetaldehyde, trans-linalool oxide, and indole were markedly lower than those in samples G and F, suggesting that these aroma substances may mask the distinctive “Cong flavor” characteristics.

### 3.4. Identification of Four Representative Active Aroma Components of LCSX via GC–O–MS

GC–O–MS analysis revealed a total of 36 active aroma components detected across the four LCSX samples. As shown in Table 2, we identified more active aroma compounds than indicated by the OAV calculations. Notably, during the olfactory evaluation, no single active aroma component was found to be consistent with the “Cong flavor”. Instead, many of the detected substances presented similar “Cong flavor” characteristics. For example, compounds such as theaspirone (woody aroma, sniffing intensity = 3), *δ*-decalactone (sweet sandalwood aroma, sniffing intensity = 2), and 2,5-dimethylpyrazine (woody aroma, sniffing intensity = 3) showed similarities. Additionally, 2,5-dimethylpyrazine (barley tea aroma and rice aroma, sniffing intensity = 1), 2-methyl-5-isopropylpyrazine (hazelnut aroma and rice aroma, sniffing intensity = 2), and 2-acetylpyrrole (rice aroma, sniffing intensity = 3) reflected rice-like characteristics, whereas coumarin (hay and Indicalamus leaf aroma, sniffing intensity = 2) resembled the aroma of the Indicalamus leaf.

Moreover, several active aroma compounds with floral, grassy, and sweet attributes, such as ethyl acetate (smiling flower aroma, sniffing intensity = 3), hexanal (grassy aroma, sniffing intensity = 4), and methyl phenylacetate (honey-like aroma, sniffing intensity = 2), were identified. These substances likely play significant roles in contributing to the “Cong flavor” of LCSX and the shared aromatic properties of oolong tea.

### 3.5. Construction of the Flavor Wheel for LCSX and Identification of Key “Cong Flavor” Compounds via GC–O–MS

Based on the GC–O results, we constructed an aroma flavor wheel for LCSX (Figure 4a). The components of the aroma profile can be categorized into two main types: The first includes those associated with “Cong flavor” attributes, including woody, Indicalamus leaf, and rice-like aromas. Within the woody category, subaromas such as refreshing herbal, almond, sweet sandalwood, and faint woody notes are included. The Indicalamus leaf aroma encompasses hints of hay, whereas the rice-like aroma includes barley tea and hazelnut characteristics. The second category consists of aromatic properties shared with oolong tea, such as floral, grassy, sweet, and fruity notes. The floral aroma category comprised sweet, intense, and subtle floral notes, with the sweet subgroup exhibiting magnolia-, michelia-, and pollen-like characteristics. The grassy category included pungent and cucumber-like attributes. Distinct fruit and sweet subcategories were identified, encompassing tropical fruit, green apple, honey, and cocoa-milk nuances. Combining the olfactory assessments and flavor wheel analysis, we conclude that the “Cong flavor” is not derived from a single compound but rather results from the synergistic interaction of multiple active aroma components.

To further identify the key active aroma substances contributing to the “Cong flavor” of LCSX, we calculated the correlations between the aroma attributes rated by the evaluation panel for the four samples and the detected concentrations (Figure 4b). The results revealed that the contents of theaspirone, dihydroactinidiolide, *δ*-decalactone, and trans-linalool oxide were significantly correlated with the woody aroma of LCSX. Additionally, 2-acetylpyrrole, 5-methylfurfural, 2,5-dimethylpyrazine, and 2-methyl-5-isopropylpyrazine were significantly correlated with the rice-like aroma, whereas coumarin was significantly associated with the Indicalamus leaf aroma. Notably, several compounds, including theaspirone, *δ*-decalactone, and 2-acetylpyrrole, align with our OAV calculations, further substantiating the contributions of these three aroma substances to the unique fragrance profile of LCSX.

### 3.6. Molecular Docking Analysis of the Binding Interactions Between “Cong Flavor” Aroma-Active Compounds and Olfactory Receptors

The lower the binding energy between a ligand and its receptor in a stable conformation is, the greater the likelihood of interaction. When the binding energy is ≤−4.25 kcal/mol (−17.78 kJ/mol), it indicates a certain level of binding activity between the active compound and the target. Binding energies ≤−5.0 kcal/mol (−20.92 kJ/mol) suggest a stronger binding affinity [29]. To investigate the roles of the three aroma-active compounds associated with “Cong favor” in oolong tea, we performed molecular docking analyses with seven human olfactory receptors to explore their contributions and potential binding sites, including human olfactory receptor 1E2 (OR1E2), human olfactory receptor 1G1 (OR1G1), human olfactory receptor 5M3 (OR5M3), human olfactory receptor 7D4 (OR7D4), human olfactory receptor 7G1 (OR7G1), human olfactory receptor 8D1 (OR8D1), and human olfactory receptor 8G1 (OR8G1).

The results, illustrated in Figure 5a, reveal that 2-acetylpyrrole acts as a small-molecule ligand and forms a stable hydrogen bond with the THR-207 residue of OR8D1. Furthermore, *δ*-decalactone and theaspirone establish hydrophobic interactions at SER-108 and HIS-164 of OR1E2, respectively (Figure 5b,c). Notably, both *δ*-decalactone and theaspirone were found to stably bind multiple olfactory receptors, each exhibiting several stable spatial conformations (Figure 5d and Appendix A). For example, *δ*-decalactone was predicted to exhibit 10 stable spatial conformations within OR1E2, OR7D4, and OR8D1. This finding provides indirect evidence for its potential molecular binding mechanisms and the intricate complexity of its interactions with olfactory receptors.

## 4. Discussion

The aroma of tea not only serves as a critical quality indicator but also significantly influences product pricing. This study employs sensory evaluation in conjunction with SBSE–GC–O–MS technology to conduct a comprehensive analysis of the key aroma components in LCSX tea samples, elucidating their aroma profiles and essential characteristic compounds. Through quantitative descriptive analysis (QDA), we identified the primary aroma characteristics of LCSX as woody, rice-like, and Indicalamus leaf notes, which aligns with the traditional concept of “Cong flavor” from Wuyi Mountain. Our analysis of four representative LCSX samples, which utilized chemometric methods and aroma activity values, enabled identification of the key aroma-active substances responsible for the unique woody, rice-like, and Indicalamus leaf aromas of LCSX, offering novel insights into the formation of its distinctive “Cong flavor”.

The results from SBSE–GC–O–MS indicate that compounds such as theaspirone, dihydroactinidiolide, *δ*-decalactone, and trans-linalool oxide are significantly correlated with the woody aroma of LCSX. Theaspirone has been shown to be widely present in green, black, and oolong teas [30]. Recent studies have identified theaspirone in tuocha, where it is a key aroma component contributing to woody and herbal notes and serves as an important indicator for assessing the aging of tuocha [31]. Dihydroactinidiolide, a terpenoid derived from *β*-carotene degradation, has woody and floral-fruity aromas [32,33]. Previous studies have shown that dihydroactinidiolide is a crucial aroma compound for the woody characteristics of Fuzhuan tea, arising from microbial degradation during tea processing [34,35]. *δ*-Decalactone is generally regarded as possessing a sweet-fatty aroma [36]. Our olfactory results indicate that the *δ*-decalactone in LCSX has a sweet sandalwood-like aroma, which differs from the literature but still falls within the sweet aroma category, potentially influenced by varying concentrations. Additionally, trans-linalool oxide not only possesses floral characteristics but also typically imparts rooty or woody notes [37,38], which is consistent with our olfactory results. 2-Acetylpyrrole was first identified in rice and is considered one of the most important aroma compounds in rice, particularly in fragrant varieties. Studies have shown that 2-acetylpyrrole has a very low odor threshold in water [39]. The low threshold makes it an important volatile compound in the formation of food aromas; in addition to rice, it is also a key flavor compound in many grain products as well as certain vegetables and animal products [40]. 5-Methylfurfural and pyrazines are widely found in various food products and are associated with nutty aromas [41,42]. Notably, 2-acetylpyrrole, 5-methylfurfural, 2,5-dimethylpyrazine, and 2-methyl-5-isopropylpyrazine are products of the Maillard reaction and are primarily formed during the tea leaf roasting process. In practical tea production, there is a notable phenomenon where an appropriate increase in roasting degree can significantly enhance the “Cong flavor”, which is likely associated with elevated levels of compounds such as 2-acetylpyrrole and 5-methylfurfural. Coumarin, a naturally occurring aromatic compound, is widely distributed throughout the plant kingdom; it is characterized by its sweet fragrance, pronounced fresh hay aroma, and nut-like flavor profile [43,44].

Our olfactory results similarly indicate that coumarin has a moderate-intensity hay-like aroma, reminiscent of the fragrance of dried bamboo leaves. Notably, during our olfactory assessments, we did not identify or detect any single aromatic compound resembling the “Cong flavor”. Therefore, we reasonably conclude that the “Cong flavor” may arise from the synergistic interaction of various woody aromatic components, resulting in a distinct fragrance profile.

## 5. Conclusions

This study systematically elucidates the chemical basis of the unique aroma of LCSX, a high-grade Wuyi rock tea characterized by its distinctive “Cong flavor”. Through an integrative approach combining sensory analysis, SBSE–GC–O–MS, and molecular sensory science, we identified key aroma-active compounds—such as theaspirone, *δ*-decalactone, and 2-acetylpyrrole—that contribute significantly to defining the woody and rice-like notes of the “Cong flavor”. Molecular docking analysis further revealed potential binding interactions between key aroma-active compounds and olfactory receptors, offering new insights into the molecular mechanisms underlying the human aroma perception of LCSX. Our findings suggest that the “Cong flavor” is not the product of a single compound but rather a synergistic interaction of multiple aroma-active substances. Future studies should incorporate larger sample sets encompassing broader geographical origins, cultivation practices, and tree age ranges to further validate these findings and explore potential correlations between agronomic factors and aroma compound biosynthesis. Such expanded investigations would provide more comprehensive insights into the complex formation mechanisms of “Cong flavor” in aged Shuixian teas.

Our results not only advance the understanding of the aroma chemistry of Laocong Shuixian but also establish a foundation for further research into the production and sensory evaluation of high-quality oolong teas. Future studies should explore the genetic and environmental factors influencing aroma compound biosynthesis in aged tea trees.

## Figures and Tables

**Figure 1 foods-14-01706-f001:**
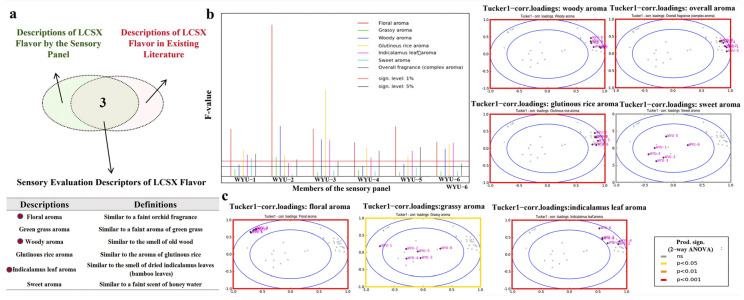
Evaluation results of sensory terms and panel performance. (**a**) Venn diagram illustrating the six consensus aroma descriptors for the “Cong flavor”. (**b**) The F value reflects the ability of the sensory panel members to distinguish between samples. (**c**) The Tucker’s value indicates the consistency of the evaluation results among the panel members. A *p* value of < 0.05 is considered indicative of consistency in the panel members’ evaluation results (the color of the outer box corresponds to different *p* value ranges).

**Figure 2 foods-14-01706-f002:**
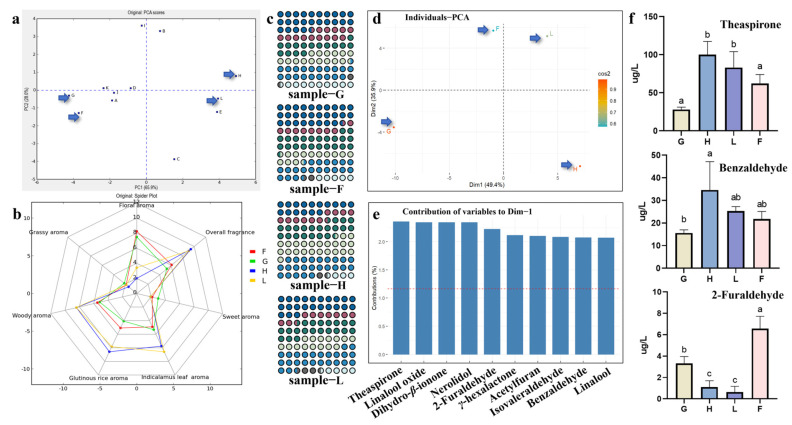
Screening and volatiles analysis of LCSX tea samples. (**a**) PCA plot based on the scoring matrix generated by the sensory evaluation panel for the 12 LCSX samples. Letters A to L were used as designations for 12 distinct LCSX tea samples. (**b**) Radar chart displaying the scoring of the six flavor sensory attributes and the overall aroma intensities of four representative LCSX tea samples, with different colors representing different samples. (**c**) Distribution map of aroma components in four representative samples of LCSX. Classification of chemical compounds represented by different colors. Blue circles correspond to alcohols, maroon circles represent aldehyde, green circles indicate ketones, light green circles represent esters, and light blue circles denote nitrogen-containing compounds. Gray circles represent phenolic compounds, while light blue circles correspond to acids. (**d**) PCA plot based on GC–MS determination of the volatile metabolic profiles of four LCSX samples. Dim1 and Dim2 represent principal component 1 and principal component 2, respectively. (**e**) Top ten volatile compounds in terms of their contribution to Dim1. (**f**) The content of three key aroma compounds in each tea sample. Bars represent mean ± SD; n = 3 biological replicates. Statistical analysis was performed by ANOVA. Means distinguished with different letters are significantly different from each other (*p* < 0.05).

**Figure 3 foods-14-01706-f003:**
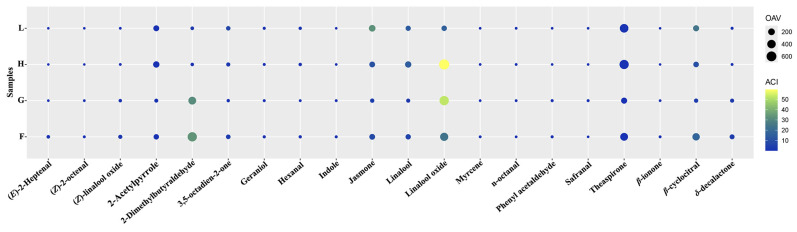
Bubble chart of the aroma-active compounds in the four LCSX tea samples. The size and color of the bubbles are used to map the OAV and ACI values of the aroma substances.

**Figure 4 foods-14-01706-f004:**
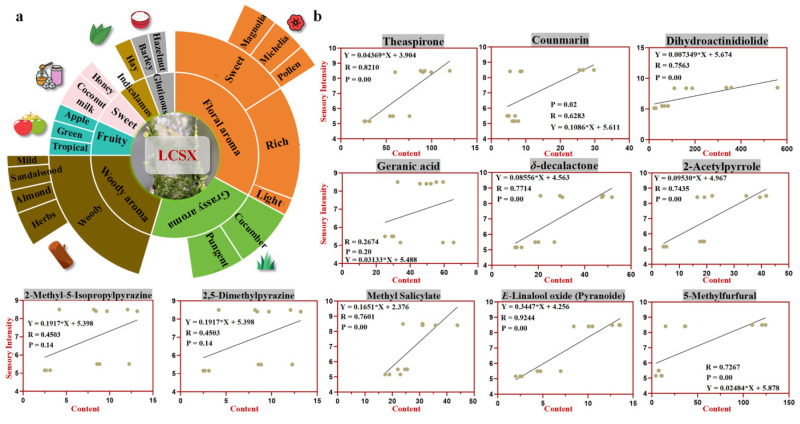
(**a**) Aroma flavor wheel of LCSX. Different color segments represent various aroma attributes. (**b**) Scatter plot illustrating the correlations between the concentrations of key aroma compounds and their corresponding sensory intensities.

**Figure 5 foods-14-01706-f005:**
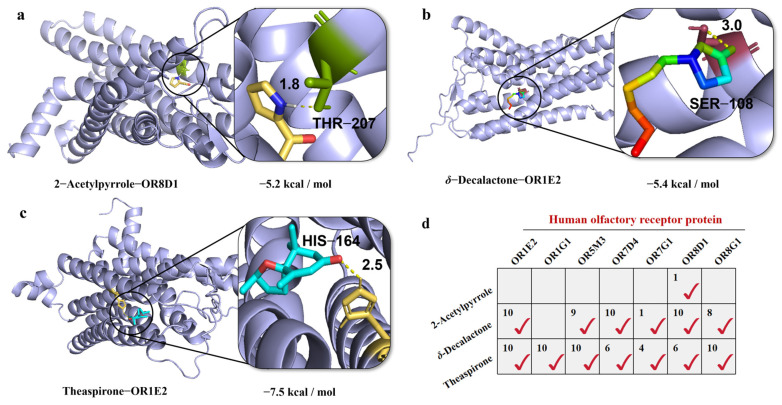
Simulation results for molecular docking with 2-acetylpyrrole (**a**), *δ*-decalactone (**b**), or theaspirone (**c**) and olfactory receptors. (**d**) The ligand–receptor binding matrix summarizes the interactions between small-molecule ligands and olfactory receptors. “√” indicates stable binding between a ligand and a receptor, and the numbers represent the distinct stable spatial conformations identified.

**Table 2 foods-14-01706-t002:** Identification of key aroma-active compounds in LCSX (ug/mL).

Compounds	RI-Lib	RI	RT	Sample G	Sample H	Sample L	Sample F	Odor Properties	Intensity
Isovaleraldehyde	925	926	7.5	4.96 ± 0.76	12.83 ± 2.27	10.22 ± 1.25	13.25 ± 1.61	Almond aroma	2
Ethyl acetate	934	942	7.9	2.39 ± 1.14	2.8 ± 0.32	1.31 ± 0.25	5.4 ± 1.2	Michelia flower-like	3
Hexanal	1094	1087	13.5	6.07 ± 1.38	40.37 ± 6.91	36.09 ± 9.27	28.43 ± 5.62	Grassy aroma	4
(*2E*)-Hexenal	1215	1231	15.4	2.6 ± 0.96	7.85 ± 2.13	7.89 ± 3.72	7.44 ± 1.95	Grassy aroma	2
2,5-Dimethylpyrazine	1336	1337	18.3	2.19 ± 1.03	8.29 ± 1.89	8.21 ± 2.15	8.58 ± 5.38	Barley tea and rice-like	1
methyl (*E*)-hex-3-enoate	1246	1265	16.3	9.71 ± 3.27	9.35 ± 2.35	11.29 ± 4.07	8.62 ± 4.71	Green apple aroma	2
2-Methyl-5-isopropylpyrazine	1394	1405	20.1	2.74 ± 0.33	8.14 ± 3.94	10.43 ± 2.47	9.88 ± 2.07	Hazelnut and rice-like	2
Linalool oxide	1470	1483	22.2	31.12 ± 12.98	38.7 ± 38.7	81.22 ± 3.09	102.58 ± 9.06	Rich floral	3
3,5-Octadien-2-one	1515	1536	23.5	3.96 ± 0.13	14.14 ± 0.12	21.26 ± 1.24	24.54 ± 1.82	Pungent grassy aroma	4
(*E,E*)-2,4-Heptadienal	1482	1513	22.9	18.21 ± 9.26	44.44 ± 6.28	51.41 ± 2.88	42.96 ± 4.71	Pungent grassy aroma	3
Benzaldehyde	1530	1549	23.8	15.61 ± 1.35	34.62 ± 12.53	25.32 ± 1.97	21.79 ± 3.32	Fresh cucumber-like	3
Linalool	1548	1553	23.9	6.14 ± 0.09	34.62 ± 4.14	17.73 ± 2.16	19.3 ± 0.66	Rich floral aroma	3
5-Methyl furfural	1573	1596	25.0	8.44 ± 3.62	118.13 ± 7.52	29.18 ± 12.41	7.63 ± 0.35	Glutinous rice aroma	1
Hotrienol	1605	1619	25.5	31.97 ± 0.96	20.32 ± 8.49	52.77 ± 5.73	62.05 ± 4.11	Light floral aroma	2
*β*-Cyclocitral	1601	1644	26.1	1.64 ± 0.36	5.48 ± 2.81	5.39 ± 0.8	5.92 ± 0.2	Sweet floral aroma	1
Phenylacetaldehyde	1630	1669	26.7	12.79 ± 6.31	18.79 ± 3.04	21.35 ± 7.95	11.26 ± 2.22	Pollen aroma-like	2
Terpineol	1698	1711	27.6	3.01 ± 1.29	18.62 ± 1.83	8.73 ± 1.6	5.69 ± 0.42	Cucumber-like	3
*γ*-Hexalactone	1703	1735	28.2	2.56 ± 1.25	8.27 ± 2.00	6.65 ± 1.77	6.82 ± 1.58	Sweet coconut milk-like	2
trans-Linalool oxide (furanoid)	1736	1754	28.6	2.37 ± 0.35	13.23 ± 0.47	9.65 ± 1.09	5.35 ± 1.4	Faint woody-like	1
Methyl phenylacetate	1758	1782	29.2	8.89 ± 1.17	6.02 ± 3.64	10.49 ± 2.75	9.02 ± 4.74	Honey-like	2
Methyl salicylate	1753	1805	29.7	19.78 ± 2.87	28.68 ± 4.14	36.91 ± 6.43	24.04 ± 1.8	Herbs and refreshing	3
Geraniol	1841	1858	30.8	18.28 ± 1.44	18.64 ± 1.43	30 ± 3.61	2.51 ± 0.34	Green fruit-like	2
*α*-Ionone	1831	1872	31.1	3.41 ± 0.28	12.75 ± 1.96	6.25 ± 1.71	4.33 ± 0.17	Sweet floral	1
Benzyl alcohol	1886	1901	31.7	2.6 ± 0.23	12.54 ± 2.67	8.12 ± 3.25	4.68 ± 1.18	Sweet floral	1
Phenylethanol	1910	1938	32.5	6.45 ± 3.76	35.6 ± 12.44	22.64 ± 4.15	15.06 ± 4.06	Sweet floral	3
*β*-Ionone	1934	1964	33.0	3.41 ± 0.28	10.49 ± 1.96	14.12 ± 4.38	26.34 ± 10.23	Tropical fruit-like	2
Jasmone	1928	1972	33.2	10.95 ± 1.13	35.29 ± 4.88	57.15 ± 18.77	35.4 ± 3.8	Intense floral	3
2-Acetylpyrrole	1975	2003	33.8	4.63 ± 0.58	32.74 ± 9.97	25.05 ± 12.53	18.4 ± 0.78	Rice aroma	3
*γ*-Nonalactone	2027	2063	35.0	3.64 ± 0.55	18.01 ± 2.45	6.64 ± 1.5	10.35 ± 2.34	Sweet coconut milk-like	2
Theaspirone	2162	2224	38.0	28.04 ± 3.12	100.01 ± 17.39	83.17 ± 20.83	62.33 ± 11.55	Woody	3
*δ*-Decalactone	2176	2235	38.1	11.17 ± 1.43	32.72 ± 13.85	43.17 ± 11.3	21.85 ± 4.44	Sweet sandalwood-like	2
Geranic acid	/	2362	40.4	52.99 ± 16.56	49.08 ± 6.08	49.1 ± 3.27	28.14 ± 2.66	Woody	3
Methyl jasmonate	2317	2367	40.5	7.55 ± 3.46	46.02 ± 11.91	30.61 ± 7.68	21.06 ± 2.13	Magnolia flower-like	3
Dihydroactinidiolide	2327	2408	41.1	24.73 ± 3.55	417.29 ± 122.71	152.78 ± 39.7	68.14 ± 13.65	Woody	4
Indole	2455	2503	45.0	67.06 ± 23.4	118.05 ± 118.05	149.4 ± 41.18	149.46 ± 35.63	Strong floral aroma	4
Coumarin	2426	2527	45.46	6.87 ± 0.87	27.29 ± 2.17	7.53 ± 1.84	5.5 ± 1.2	Hay/Indicalamus leaf-like	2

RI-Lib represents the retention index of the compound from the reference database; RI indicates the experimentally determined retention index of the compound in our study; and RT stands for retention time.

## Data Availability

The original contributions presented in this study are included in the article/Appendix A. Further inquiries can be directed to the corresponding authors.

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
