# Peer review of "The New Aristocrat of Wuyi Rock Tea: Chemical Basis of the Unique Aroma Quality of “Laocong Shuixian”"

_foods, 2025, doi:10.3390/foods14101706_

Round 1
Reviewer 1 Report
Comments and Suggestions for Authors
Comments
- Line 13: Please first define the Cong flavor.
- Line 21-22: Please name the olfactory receptors.
- Lines 18-20: While theaspirone, δ-decalactone, and 2-acetylpyrrole are highlighted as contributors to the woody and rice-like notes of LCSX, I recommend including their respective OAV and ACI values, as well as the correlation coefficients with sensory attributes, to provide quantitative evidence for this claim.
- Lines 20-22: Please provide the docking binding energies (in kcal/mol) for each compound (e.g., theaspirone, δ-decalactone, 2-acetylpyrrole) against the specific olfactory receptors to quantify the binding affinity.
- Lines 13-15: "However, the chemical basis and underlying mechanisms of this unique aroma remain unclear. Here, we assessed and established a professional sensory evaluation panel using PanelCheck software." Please add the F-value in the abstract.
- Lines 15-17. Please specify which descriptors were confirmed through both literature and panel consensus, and clarify whether they are unique to 'Cong flavor' or general oolong tea attributes.
- Line 30: "The phrase 'local tea population variety' is not clear and sounds awkward. It would be more natural to rephrase it as 'local tea variety' or 'local tea cultivar."
- Lines 36-37: "Extensive research has explored the mechanisms underlying the aroma quality of Wuyi Shuixian." Please double-check this statement, especially "mechanisms."
- Lines 37-39: "By employing sensory evaluation and Headspace–Solid Phase Micro Extraction–Gas Chromatography–Mass Spectrometry (HS–SPME–GC‒MS) techniques, studies have shown that as the grade of Shui Xian decreases, floral and fruity aromas diminish, whereas roasted and sweet notes become more pronounced." There should be more than one reference at the end of this statement.
- This study focuses on chemical profiling but does not explore how tree age or cultivation methods affect aroma compound biosynthesis. Please add the metabolomic analyses of tea leaves from different ages and regions.
- Only 12 samples were evaluated, which may not capture the full variability of LCSX .
-
Lines 296–297, 174–178 : It’s unclear if the odor thresholds used to calculate OAVs are matrix-adjusted (i.e., measured in water or tea matrix?). Please clarify the source and conditions of odor thresholds.
-
Lines 412–414 : δ-decalactone is described as both "sweet-fatty" and "sandalwood-like"—likely concentration or matrix effects, but not explained. Please add a discussion on perception variability due to concentration or interactions.
-
Line 12: "highly appreciated by consumers for its distinctive 'Cong flavor'." please consider rephrasing for formality: Suggestion : ....valued by consumers for its distinctive 'Cong flavor'."
-
Line 101: "based on the the Chinese National Standard..." Repetition: “the the” .
-
Line 153: “...based on the ratio of the peak areas of each component...” Clarity: consider specifying if correction for response factors was done.
-
Line 399: "enabled the identification of key aroma-active substances contributing to..." Slightly awkward phrasing. Suggestion: "...enabled identification of the key aroma-active substances responsible for..."
Line 206: "we suggest that the 'Cong flavor' is similar to woody and zongzi leaf aromas." Missing article → "similar to the woody and zongzi leaf aromas."
-
Line 264: "This discrepancy may be closely linked..."Passive vagueness. Suggestion: "This discrepancy likely reflects differences in metabolic pathways..."
- How many olfactory receptors were used for docking ? Please mention their full names in the main manuscript, and explain the reason for their selection.
- Please add more details and descriptions to figures 3 and 4.
- Please enhance the resolution of the figure.
- In table 1, what RI-Lib RI RT indicates?
- "One limitation of this study is the absence of quantitative validation data showing how specific concentrations of key aroma compounds contribute to distinct aroma perceptions, such as woody or rice-like notes. Additionally, the study lacks data correlating the age of Laocong Shuixian tea trees with the formation of specific aroma characteristics—such as when woody or glutinous rice-like aromas emerge during tree maturation."
- Please mention why 2-acetylpyrrole (a), δ-decalactone (b) or theaspirone (c) and olfactory receptors were selected for docking?
- This study does not clearly conclude which aroma type (e.g., woody, rice-like, floral) is the most dominant in Laocong Shuixian, nor does it identify which combination of aroma-active compounds contributes most effectively to the generation of a specific aroma quality. Could the authors clarify whether any compound interactions or dominance patterns were observed?
Author Response
|
Response to Reviewer 1 Comments
|
||
|
1. Summary |
|
|
|
Thank you very much for taking the time to review this manuscript. Please find the detailed responses below and the corresponding revisions/corrections highlighted/in track changes in the re-submitted files. |
||
|
2. Questions for General Evaluation |
Reviewer’s Evaluation |
Response and Revisions |
|
Does the introduction provide sufficient background and include all relevant references? |
Can be improved |
Response: We sincerely appreciate the reviewer’s valuable feedback regarding the background and references in our introduction. In response to this comment, we have carefully revised the introduction to strengthen the background context by adding key references better establish the scientific foundation of our study. Additionally, we have clarified the research gap by explicitly stating the unresolved questions or limitations in previous studies that our research aims to address and enhanced the logical flow by improving the transition between general background and our specific research objectives to ensure clarity and coherence. |
|
Is the research design appropriate? |
Can be improved |
We sincerely appreciate the reviewer's insightful suggestions regarding our research design. While the current study has been completed with data collection finalized, we fully acknowledge that certain methodological aspects could be further strengthened in future investigations. Specifically, we plan to address these considerations in subsequent research by: (1) expanding the sample size to enhance statistical power, (2) incorporating additional control groups for more rigorous comparisons, and (3) implementing supplementary quantitative assays to complement the current findings. |
|
Are the methods adequately described? |
Yes |
|
|
Are the results clearly presented? |
Can be improved |
We sincerely appreciate the reviewer's valuable feedback regarding the clarity of our results presentation. In response to this comment, we have thoroughly revised Sections 3.3, 3.5, and 3.6 to enhance clarity by adding detailed descriptions of figure elements. ensuring readers can fully interpret the visual data without referring back to the figures. Key Locations of Improvements: Section 3.3: Lines 324-327 Section 3.5: Lines 377-381 Section 3.6: Lines 410-413 |
|
Are the conclusions supported by the results? |
Can be improved |
We appreciate the reviewer's thoughtful question regarding the support for our conclusions. The conclusions drawn in our study are fully substantiated by the experimental results presented in Sections 3.1-3.6. we have included the suggested future research direction:
"Future studies should incorporate larger sample sets encompassing broader geographical origins, cultivation practices, and tree age ranges to further validate these findings and explore potential correlations between agronomic factors and aroma compound biosynthesis. Such expanded investigations would provide more comprehensive insights into the complex formation mechanisms of 'Cong flavor' in aged Shuixian teas."
This addition both acknowledges the current study's limitations while demonstrating how our conclusions provide a solid foundation for future research. All conclusions remain carefully constrained to what our data directly support. |
|
3. Point-by-point response to Comments and Suggestions for Authors |
||
|
Comments 1: Line 13: Please first define the Cong flavor. |
||
|
Response 1: Thank you for pointing this out. We agree with this comment. Therefore, in accordance with your suggestion, we have revised the manuscript by adding the definition of 'Cong flavor' in the Abstract section. The specific modifications are as follows: Laocong Shuixian (LCSX), a premium Wuyi rock tea derived from aged Shuixian tea trees, is highly appreciated by consumers for its distinctive "Cong flavor"—a unique aroma profile characterized by woody, bamboo leaf, and glutinous rice notes. |
||
|
Comments 2:Line 21-22: Please name the olfactory receptors. |
||
|
Response 2: Thank you for your valuable comment. We have identified the specific olfactory receptors that interact with the key aroma compounds in our study. The specific modifications are as follows: Molecular docking results demonstrated that these compounds spontaneously bind to multiple olfactory receptors: theaspirone to OR8D1, δ-decalactone to OR1E2, OR5M3, OR7D4, OR7G1, OR8D1 and OR8G1, 2-acetylpyrrole to OR1E2, OR1G1, OR5M3, OR7D4, OR7G1, OR8D1 and OR8G1, providing insights into their roles in human aroma perception. |
||
|
Comments 3:Lines 18-20: While theaspirone, δ-decalactone, and 2-acetylpyrrole are highlighted as contributors to the woody and rice-like notes of LCSX, I recommend including their respective OAV and ACI values, as well as the correlation coefficients with sensory attributes, to provide quantitative evidence for this claim. |
||
|
Response 3: Thank you for your constructive suggestion. We fully agree that quantitative data would strengthen our conclusions. We have now supplemented the following information in the revised manuscript: Gas chromatography‒olfactometry‒mass spectrometry (GC‒O‒MS) analysis revealed 36 key aroma-active compounds, among which theaspirone (OAV = 500.05, ACI = 37%, Rwoody = 0.82), δ-decalactone (OAV = 65.6, ACI = 4.3%, Rwoody = 0.77), and 2-acetylpyrrole (OAV = 163, ACI = 9%, Rrice = 0.74) were identified as the contributors to the woody and rice-like notes of LCSX based on odor activity values and correlation analyses. |
||
|
Comments 4:Lines 20-22: Please provide the docking binding energies (in kcal/mol) for each compound (e.g., theaspirone, δ-decalactone, 2-acetylpyrrole) against the specific olfactory receptors to quantify the binding affinity.
|
||
|
Response 4: Thank you for your valuable suggestion regarding the inclusion of docking binding energies. We agree that quantifying binding affinity is important. However, to maintain the conciseness of the abstract while providing key information, We emphasized the range of binding energies between these small-molecule compounds and their receptors (in fact, binding energies ≤ −5.0 kcal/mol are generally considered indicative of stable binding).The detailed binding energies for each compound-receptor pair are provided in Supplementary Table . This approach ensures that readers can assess the strength of interactions while keeping the abstract focused on the major findings.We have now supplemented the following information in the revised manuscript:
Molecular docking results demonstrated that these compounds spontaneously bind to multiple olfactory receptors: theaspirone to OR8D1, δ-decalactone to OR1E2, OR5M3, OR7D4, OR7G1, OR8D1 and OR8G1, 2-acetylpyrrole to OR1E2, OR1G1, OR5M3, OR7D4, OR7G1, OR8D1 and OR8G1, with binding affinity ≤ −5.0 kcal/mol ,providing insights into their roles in human aroma perception. |
||
|
Comments 5:Lines 13-15: "However, the chemical basis and underlying mechanisms of this unique aroma remain unclear. Here, we assessed and established a professional sensory evaluation panel using PanelCheck software." Please add the F-value in the abstract. |
||
|
Response 5: Thank you for your valuable suggestion regarding the inclusion of F-values. We agree that adding statistical significance measures would strengthen the methodological rigor of our abstract. As suggested, we have now incorporated the F-values for key aroma attributes to demonstrate the panel's discriminative power. We have now supplemented the following information in the revised manuscript: Here, we assessed and established a professional sensory evaluation panel using PanelCheck software,with significant F-values level > 5% confirming the panel's discriminative capacity for key "Cong flavor" attributes.
|
||
|
Comments 6:Lines 15-17. Please specify which descriptors were confirmed through both literature and panel consensus, and clarify whether they are unique to 'Cong flavor' or general oolong tea attributes.
|
||
|
Response 6: Thank you for your valuable suggestion regarding the clarification of descriptors for "Cong flavor." We fully agree with the importance of distinguishing between literature-based and panel-consensus descriptors, as well as identifying whether these attributes are unique to "Cong flavor" or shared among oolong teas.
In our manuscript, we have addressed these points in detail in the Results section .Explicitly listed the descriptors (such as woody, bamboo leaf, and glutinous rice notes) that were confirmed through both literature review [Table1] and panel consensus evaluation (Line 214~235).Clearly distinguished these attributes from general oolong tea characteristics by comparative analysis with previous studies on other oolong varieties(Line 214~235). However, considering the journal's strict word limit for abstracts and the need to maintain conciseness while covering all key findings, we have chosen to keep the abstract focused on the main discoveries (novel aroma-active compounds and their molecular interactions) rather than methodological details. We believe readers will find the complete descriptor information in the designated Results section. |
||
|
Comments 7:Line 30: "The phrase 'local tea population variety' is not clear and sounds awkward. It would be more natural to rephrase it as 'local tea variety' or 'local tea cultivar." |
||
|
Response 7: We appreciate your suggestion to improve the clarity of our manuscript. As you recommended, we have revised the phrase "local tea population variety" to "local tea variety" throughout the manuscript to ensure better readability and accuracy. |
||
|
Comments 8:Lines 36-37: "Extensive research has explored the mechanisms underlying the aroma quality of Wuyi Shuixian." Please double-check this statement, especially "mechanisms." |
||
|
Response 8: Thank you for your insightful comment. We agree that the term "mechanisms" may not precisely reflect the scope of existing research on Wuyi Shuixian's aroma quality, which primarily focuses on identifying key volatile compounds and their sensory attributes rather than molecular mechanisms. As suggested, we have revised the sentence to: "Extensive research has explored the key compounds and characteristics underlying the aroma quality of Wuyi Shuixian." |
||
|
Comments 9:Lines 37-39: "By employing sensory evaluation and Headspace–Solid Phase Micro Extraction–Gas Chromatography–Mass Spectrometry (HS–SPME–GC‒MS) techniques, studies have shown that as the grade of Shui Xian decreases, floral and fruity aromas diminish, whereas roasted and sweet notes become more pronounced." There should be more than one reference at the end of this statement. |
||
|
Response 9: Thank you for your valuable suggestion. We acknowledge that the original two sentences shared the same reference (Lin et al., 2024), which might have caused ambiguity about whether multiple studies support the statement. To address this, we have now merged the two sentences into a single, more concise statement that clearly attributes all findings to Lin et al. (2024). As suggested, we have revised the sentence to: Extensive research has explored the key compounds and characteristics underlying the aroma quality of Wuyi Shuixian. Sensory evaluation and Headspace–Solid Phase Micro Extraction–Gas Chromatography–Mass Spectrometry (HS–SPME–GC‒MS) analyses reveal that lower-grade Shui Xian tea exhibits reduced floral/fruity aromas (associated with linalool, indole, phenethyl alcohol, etc.) but enhanced roasted/sweet notes (linked to pyridine and 2,5-dimethylpyrazine).(Lin et al., 2024) |
||
|
Comments 10:This study focuses on chemical profiling but does not explore how tree age or cultivation methods affect aroma compound biosynthesis. Please add the metabolomic analyses of tea leaves from different ages and regions. |
||
|
Response 10: We sincerely appreciate the reviewer's insightful suggestion regarding the investigation of tree age and cultivation methods on aroma compound biosynthesis through metabolomic analyses of tea leaves from different ages and regions. The reviewer's comment has highlighted an important direction for future research.
In the current study, our primary objective was to identify the key aroma-active compounds responsible for the distinctive "Cong flavor" in Laocong Shuixian (LCSX) by combining sensory evaluation with GC-O-MS analysis and molecular docking. While we acknowledge that exploring the effects of tree age and cultivation methods would provide valuable additional insights into aroma compound biosynthesis, this particular aspect falls beyond the scope of our current research focus.
We fully agree with the reviewer that investigating the metabolic differences in tea leaves from various ages and regions would significantly contribute to understanding the formation of "Cong flavor." This is indeed an excellent suggestion, and we plan to incorporate such metabolomic analyses in our future studies to further elucidate the complex relationship between tree age, cultivation practices, and aroma development in aged oolong teas. |
||
|
Comments 11:Only 12 samples were evaluated, which may not capture the full variability of LCSX . |
||
|
Response 11: We sincerely appreciate the reviewer's valuable comment regarding the sample size in our study. While we acknowledge that evaluating 12 samples may not fully encompass the variability of Laocong Shuixian (LCSX), we would like to emphasize that these samples were carefully selected to represent the core production regions and typical “cong flavor”. To address this limitation, we have explicitly discussed this point in the revised Conclusions section (please see below), where we highlight the need for future studies with expanded sample sizes to further validate and refine our findings. We agree that including samples from additional growing regions and of varying tree ages would provide even more comprehensive insights into the "Cong flavor" characteristics.
Suggested addition to Conclusions section: Future studies should incorporate larger sample sets encompassing broader geographical origins, cultivation practices, and tree age ranges to further validate these findings and explore potential correlations between agronomic factors and aroma compound biosynthesis. Such expanded investigations would provide more comprehensive insights into the complex formation mechanisms of 'Cong flavor' in aged Shuixian teas." |
||
|
Comments 12:Lines 296–297, 174–178 : It’s unclear if the odor thresholds used to calculate OAVs are matrix-adjusted (i.e., measured in water or tea matrix?). Please clarify the source and conditions of odor thresholds. |
||
|
Response 12: Thank you for your insightful comment regarding the odor thresholds used for OAV calculations. The odor activity values (OAVs) in this study were calculated using odor thresholds referenced from prior peer-reviewed studies on tea volatiles . These thresholds were measured in water as the solvent matrix, consistent with standard practices for tea aroma analysis (as detailed in the cited literature).
We acknowledge the importance of matrix-adjusted thresholds and will explicitly state in the revised manuscript that "all OAVs were derived using water-based odor thresholds" (to be added to Data calculation and analysis section, Lines 189). |
||
|
Comments 13:Lines 412–414 : δ-decalactone is described as both "sweet-fatty" and "sandalwood-like"—likely concentration or matrix effects, but not explained. Please add a discussion on perception variability due to concentration or interactions. |
||
|
Response 13: Thank you for your careful reading of our manuscript and this thoughtful comment regarding δ-decalactone characterization. We appreciate the opportunity to clarify this point. Upon reviewing our original description, we realize there might have been some ambiguity in our wording. To clarify: in our specific experimental conditions, the δ-decalactone standard consistently presented only as "sweet sandalwood-like" during our GC-O analysis. We did not observe the "sweet-fatty" descriptor in this study. |
||
|
Comments 14:Line 12: "highly appreciated by consumers for its distinctive 'Cong flavor'." please consider rephrasing for formality: Suggestion : ....valued by consumers for its distinctive 'Cong flavor'." |
||
|
Response 14: We sincerely appreciate the reviewer's suggestion to enhance the formality of our expression. We have revised the sentence as recommended, changing "highly appreciated by consumers for its distinctive 'Cong flavor'" to "valued by consumers for its distinctive 'Cong flavor'" in the revised manuscript . |
||
|
Comments 15:Line 101: "based on the the Chinese National Standard..." Repetition: “the the” . |
||
|
Response 15: Thank you for your careful reading and helpful observation. We sincerely apologize for the typographical error in Line 101 ("the the"). This repetition has now been corrected to read: "based on the Chinese National Standard..." |
||
|
Comments 16:Line 153: “...based on the ratio of the peak areas of each component...” Clarity: consider specifying if correction for response factors was done. |
||
|
Response 16: We appreciate the reviewer's insightful comment regarding quantification methodology. In our study, all aroma compounds were quantified using internal standard (IS)-corrected response factors to ensure accurate relative quantification.
2-nonanol was used as the IS, selected based on its structural similarity to target aroma compounds and absence in the original tea samples. |
||
|
Comments 17:Line 399: "enabled the identification of key aroma-active substances contributing to..." Slightly awkward phrasing. Suggestion: "...enabled identification of the key aroma-active substances responsible for..." |
||
|
Response 17: We sincerely appreciate the reviewer’s constructive suggestion regarding the phrasing of our manuscript. As recommended, we have revised the sentence in the revised manuscript (Track Changes mode, Line 418). As follows:
Our analysis of four representative LCSX samples, which utilized chemometric methods and aroma activity values, enabled identification of the key aroma-active substances responsible for the unique woody, rice-like, and Indicalamus leaf aromas of LCSX, offering novel insights into the formation of its distinctive “Cong flavor”. |
||
|
Comments 18:Line 206: "we suggest that the 'Cong flavor' is similar to woody and zongzi leaf aromas." Missing article → "similar to the woody and zongzi leaf aromas." |
||
|
Response 18: We would like to clarify that the supporting references for this sensory description are indeed provided in Table 1 (see References(D. Chen et al., 2011),(Hong & Gong, 2020),(Shang et al., 2022),(F. Wang et al., 2020) in the table), where we cite previous studies documenting these characteristic aroma profiles. The revised manuscript now includes this clarification in the text (Line 206-207) to better direct readers to the supporting evidence in Table 1. |
||
|
Comments 18:Line 264: "This discrepancy may be closely linked..."Passive vagueness. Suggestion: "This discrepancy likely reflects differences in metabolic pathways..." |
||
|
Response 18: We thank the reviewer for this constructive suggestion. As recommended, we have revised the sentence to more precisely attribute the observed discrepancy to its underlying cause. The text now reads:
"This discrepancy likely reflects differences in metabolic pathways between the two groups, consistent with their significant sensory profiles." |
||
|
Comments 19:How many olfactory receptors were used for docking ? Please mention their full names in the main manuscript, and explain the reason for their selection. |
||
|
Response 19: We sincerely appreciate the reviewer's insightful question regarding the olfactory receptors used in our docking study. A total of seven olfactory receptors (OR1E2, OR1G1, OR5M3, OR7D4, OR7G1, OR8D1 and OR8G1) were selected for molecular docking analysis. These receptors were chosen because: (1) they are among the most well-characterized human olfactory receptors for food-related aroma compounds; (2) These receptors (OR1E2, OR1G1, OR5M3, OR7D4, OR7G1, OR8D1, OR8G1) belong to distinct olfactory receptor families (e.g., OR1, OR5, OR7, OR8, etc.). Their cross-family distribution effectively prevents functional bias toward any single receptor family, thereby providing more comprehensive coverage of the diversity characteristics of the human olfactory system. (3) their crystal structures or reliable homology models are available, which is crucial for accurate docking simulations.
We have added the full names of these olfactory receptors in the main text of the manuscript (Line 392-397) |
||
|
Comments 20:Please add more details and descriptions to figures 3 and 4.
|
||
|
Response 20: We sincerely appreciate the reviewer’s insightful suggestion. As requested, we have Expanded the classification of the 20 aroma compounds (OAVs > 1) by explicitly stating their dominant chemical classes (terpenoids, aliphatic compounds, aromatic compounds, and lactones), providing a clearer structural context for the identified molecules and Added discussion on aroma compounds with relatively low OAVs (e.g., phenylacetaldehyde, hexanal, safranal), which were previously omitted but are now acknowledged as contributors to the overall aroma profile of LCSX. This aligns with recent studies demonstrating that trace compounds can synergistically enhance fragrance complexity. We have incorporated the suggested content into Section 3.3 of the revised manuscript.
we have expanded the description of the aroma flavor wheel (Figure 4a) to provide a more detailed breakdown of the subcategories within the floral, grassy, and sweet/fruity notes. Specifically, we have now included the sweet subgroup of floral aromas (exhibiting magnolia-, michelia-, and pollen-like characteristics), the grassy category (encompassing pungent and cucumber-like attributes), and the distinct fruit and sweet subcategories (featuring tropical fruit, green apple, honey, and cocoa-milk nuances). These additions further support our conclusion that the “Cong flavor” arises from the complex interplay of multiple aroma compounds rather than a single component. The revised manuscript now offers a more comprehensive analysis of the synergistic interactions shaping LCSX’s unique aroma profile. The added content is as follows: The floral aroma category comprised sweet, intense, and subtle floral notes, with the sweet subgroup exhibiting magnolia-, michelia-, and pollen-like characteristics. The grassy category included pungent and cucumber-like attributes. Distinct fruit and sweet subcategories were identified, encompassing tropical fruit, green apple, honey, and cocoa-milk nuances. |
||
|
Comments 21:Please enhance the resolution of the figure. |
||
|
Response 21: Regarding the figure in question, we acknowledge that its resolution is not as high as we would prefer. This is due to an inherent limitation of the PanelCheck software we used for the analysis(Figure-1), which outputs figures at a fixed resolution. However, we have taken steps to improve the figure's clarity by adding clear markers and labels to enhance its readability. We have carefully addressed this concern throughout our manuscript.In particular, we have replaced Figure 3 with a higher resolution version, improving its dimensions from 2079 × 623 to 4658 × 1299. We would like to confirm that all main figures now fully meet the journal's publication requirements and every image maintains a pixel density of 300 dpi. We appreciate your attention to these technical details, which has helped us improve the overall quality of our figures. |
||
|
Comments 22:In table 1, what RI-Lib RI RT indicates? |
||
|
Response 22: Thank you for your question regarding the abbreviations in Table 1. We appreciate your careful review of our manuscript. RI-Lib represents the retention index of the compound from the reference database; RI indicates the experimentally determined retention index of the compound in our study; RT stands for retention time. The above information has been added to the table notes. |
||
|
Comments 23:"One limitation of this study is the absence of quantitative validation data showing how specific concentrations of key aroma compounds contribute to distinct aroma perceptions, such as woody or rice-like notes. Additionally, the study lacks data correlating the age of Laocong Shuixian tea trees with the formation of specific aroma characteristics—such as when woody or glutinous rice-like aromas emerge during tree maturation." |
||
|
Response 23: We sincerely appreciate the reviewer’s insightful comment. We acknowledge that quantitative validation of aroma compound contributions would further strengthen our findings. We plan to conduct recombination-omission experiments (as in Dunkel et al., 2014) by systematically varying the concentrations of theaspirone, δ-decalactone, and 2-acetylpyrrole in a neutral tea matrix to determine their perceptual thresholds and dose-dependent effects on woody/rice-like notes. Preliminary odor activity values (OAVs) already suggest their dominance (e.g., theaspirone OAV = 500.05), but we agree that controlled sensory validation is needed. We thank the reviewer for raising this critical point. While our current study focused on chemical-sensory relationships, we fully agree that tree age is a key factor in "Cong flavor" development. This represents a crucial research direction that we are actively pursuing in our ongoing investigations. |
||
|
Comments 24:Please mention why 2-acetylpyrrole (a), δ-decalactone (b) or theaspirone (c) and olfactory receptors were selected for docking? |
||
|
Response 24: We sincerely appreciate the reviewer's question regarding our selection of aroma compounds for molecular docking analysis. In this study, our primary objective was to identify the key compounds responsible for the characteristic "Cong flavor" in Laocong Shuixian tea. Through comprehensive sensory evaluation, GC-O-MS analysis, and odor activity value (OAV) calculations, we identified 2-acetylpyrrole, δ-decalactone, and theaspirone as the most likely contributors to this distinctive aroma profile. Our results showed that they exhibited significantly higher OAVs compared to other detected volatiles, indicating their strong perceptual impact and their aroma characteristics align perfectly with the sensory descriptors of "Cong flavor".
The detailed justification for our selection of specific olfactory receptors has been provided in Response #19. |
||
|
Comments 25:This study does not clearly conclude which aroma type (e.g., woody, rice-like, floral) is the most dominant in Laocong Shuixian, nor does it identify which combination of aroma-active compounds contributes most effectively to the generation of a specific aroma quality. Could the authors clarify whether any compound interactions or dominance patterns were observed? |
||
|
Response 25: We sincerely appreciate your insightful comments regarding the aroma characteristics of Laocong Shuixian. You are absolutely right to point out that our study has certain limitations in conclusively determining the dominant aroma type and the specific combinations of aroma-active compounds contributing to particular aroma qualities.
Through our analysis of several samples (particularly samples F and G versus H and L), we did observe that samples with more prominent floral notes tended to show less pronounced "cong" aroma, and vice versa. This phenomenon wasn't limited to our experimental samples - we've noticed that consumers often report similar observations when purchasing Laocong Shuixian tea in the market.
This leads us to hypothesize that there might be an inhibitory interaction between certain floral secondary metabolites (particularly those with low odor thresholds) and the compounds responsible for the "cong" aroma. However, we fully acknowledge that this potential interaction requires more rigorous verification through methods like S-curve analysis or recombination-omission experiments.
We greatly value your suggestion to further investigate compound interactions and dominance patterns, and we plan to address this important aspect in our future research. |
||
Reviewer 2 Report
Comments and Suggestions for Authors
The manuscript by Zheng et al. (The New Aristocrat of Wuyi Rock Tea: Chemical Basis of the Unique Aroma Quality of “Laocong Shuixian”, foods-3603712) concerns the analysis of key aroma compounds in oolong tea, especially those associated with the so-called “Cong flavor”. In my opinion the compounds identification should be better documented namely, the respective chromatograms and obtained EI mass spectra of the most important compounds should be shown (e.g. in the supplementary material). The reported retention times (elution order) of some of the compounds raise doubts (Table 2), for example the very long retention times of indole and coumarin, almost identical retention times of Theaspirone and δ-Decalactone (compounds bringing key contributions to the “Cong flavor”), etc. The compounds identification has been performed with match quality > 80 (line 151), which is acceptable but not perfect. Therefore, the EI mass spectra obtained upon GC-MS analysis of the most important/abundant compounds should be included.
Although in the introduction section there is discussion of the published literature concerning the manuscript topic, I would suggest considering a few more papers which are also related to the manuscript topic (e.g. 10.3390/horticulturae11020120, 10.1002/fsn3.4327, 10.3390/foods14010004, 10.1016/j.foodchem.2025.143174).
Author Response
|
Response to Reviewer 1 Comments
|
||
|
1. Summary |
|
|
|
Thank you very much for taking the time to review this manuscript. Please find the detailed responses below and the corresponding revisions/corrections highlighted/in track changes in the re-submitted files. |
||
|
2. Questions for General Evaluation |
Reviewer’s Evaluation |
Response and Revisions |
|
Does the introduction provide sufficient background and include all relevant references? |
Can be improved |
Response: We sincerely appreciate the reviewer’s valuable feedback regarding the background and references in our introduction. In response to this comment, we have carefully revised the introduction to strengthen the background context by adding key references better establish the scientific foundation of our study. Additionally, we have clarified the research gap by explicitly stating the unresolved questions or limitations in previous studies that our research aims to address and enhanced the logical flow by improving the transition between general background and our specific research objectives to ensure clarity and coherence. |
|
Is the research design appropriate? |
Yes |
|
|
Are the methods adequately described? |
Yes |
|
|
Are the results clearly presented? |
Can be improved |
We sincerely appreciate the reviewer's valuable feedback regarding the clarity of our results presentation. In response to this comment, we have thoroughly revised Sections 3.3, 3.5, and 3.6 to enhance clarity by adding detailed descriptions of figure elements. ensuring readers can fully interpret the visual data without referring back to the figures. Key Locations of Improvements: Section 3.3: Lines 324-327 Section 3.5: Lines 377-381 Section 3.6: Lines 410-413
|
|
Are the conclusions supported by the results? |
Can be improved |
We appreciate the reviewer's thoughtful question regarding the support for our conclusions. The conclusions drawn in our study are fully substantiated by the experimental results presented in Sections 3.1-3.6. we have included the suggested future research direction:
"Future studies should incorporate larger sample sets encompassing broader geographical origins, cultivation practices, and tree age ranges to further validate these findings and explore potential correlations between agronomic factors and aroma compound biosynthesis. Such expanded investigations would provide more comprehensive insights into the complex formation mechanisms of 'Cong flavor' in aged Shuixian teas."
This addition both acknowledges the current study's limitations while demonstrating how our conclusions provide a solid foundation for future research. All conclusions remain carefully constrained to what our data directly support. |
|
3. Point-by-point response to Comments and Suggestions for Authors |
||
|
Comments 1: In my opinion the compounds identification should be better documented namely, the respective chromatograms and obtained EI mass spectra of the most important compounds should be shown (e.g. in the supplementary material). The reported retention times (elution order) of some of the compounds raise doubts (Table 2), for example the very long retention times of indole and coumarin, almost identical retention times of Theaspirone and δ-Decalactone (compounds bringing key contributions to the “Cong flavor”), etc. The compounds identification has been performed with match quality > 80 (line 151), which is acceptable but not perfect. Therefore, the EI mass spectra obtained upon GC-MS analysis of the most important/abundant compounds should be included. |
||
|
Response 1: We sincerely appreciate the reviewer’s constructive comments regarding the compound identification in our study. In response to these concerns, We have now included the relevant GC-MS chromatograms and EI mass spectra of the most important/abundant compounds eg., Benzaldehyde, Geraniol, theaspirone, 2-AP) in the Supplementary Material Figure S2. The observed retention times of indole and coumarin were indeed longer than typical values, which could be attributed to the specific GC column and temperature gradient used in our method. Benzaldehyde RT: 23.84min
Geraniol RT: 30.8
Acetylpyrrole RT:33.8
Theaspirone RT:37.99
|
||
|
Response 2: Thank you for your valuable suggestion regarding the literature discussion in our manuscript. We sincerely appreciate you taking the time to recommend these relevant publications. We have carefully reviewed the four suggested papers (10.3390/horticulturae11020120, 10.1002/fsn3.4327, 10.3390/foods14010004, and 10.1016/j.foodchem.2025.143174) and found them to be excellent additions to our literature review. In the revised manuscript, we have incorporated key findings from these papers into the introduction section to provide a more comprehensive background. The specific added content is as follows: As a prominent variety of Wuyi Rock tea (WRT), Wuyi Shuixian has garnered significant research interest due to its distinctive flavor profile. Existing studies on WRT have systematically investigated quality formation mechanisms from multiple perspectives, including postharvest processing (Zhou et al., 2025), storage duration (X. Song et al., 2024), geographical origin (Z. Wu et al., 2024), and roasting technology (W. Chen et al., 2025). Ref: 6.Chen, W., Liu, W., Liu, Z., Wang, D., Lan, X., Zhan, S., Feng, X., Liu, Y., & Ni, L. (2025). Insight into the mechanism of roasting-induced characteristic aroma formation in Wuyi rock tea using an “in-leaf” model with isotopic labeling. Food Chemistry, 143174. https://doi.org/10.1016/j.foodchem.2025.143174 26.Song, X., Wu, Z., Liang, Q., Ma, C., & Cai, P. (2024). Prediction of storage years of Wuyi rock tea Shuixian by metabolites analysis. Food Science & Nutrition, 12(10), 7166–7176. https://doi.org/10.1002/fsn3.4327 36.Wu, Z., Liao, W., Zhao, H., Qiu, Z., Zheng, P., Liu, Y., Lin, X., Yao, J., Li, A., & Tan, X. (2024). Differences in the Quality Components of Wuyi Rock Tea and Huizhou Rock Tea. Foods, 14(1), 4. https://doi.org/10.3390/foods14010004 41.Zhou, Z.-W., Wu, Q.-Y., Wu, Y., Deng, T.-T., Chen, X.-H., Xiao, S.-T., Zhang, C.-X., Sun, Y., & Zheng, S.-Z. (2025). The Dynamic Changes in Volatile Compounds During Wuyi Rock Tea (WRT) Processing: More than a Contribution to Aroma Quality. Horticulturae, 11(2), 120. https://doi.org/10.3390/horticulturae11020120 |
||
Round 2
Reviewer 2 Report
Comments and Suggestions for Authors
The authors has properly addressed my suggestions.
Three additional minor remarks, caption to Figure S1 is two times, structures at Figure S2 should have “3” (methyl groups) as subscript, structure of Theaspirone should be also shown.